# Heteronemin and Tetrac Induce Anti-Proliferation by Blocking EGFR-Mediated Signaling in Colorectal Cancer Cells

**DOI:** 10.3390/md20080482

**Published:** 2022-07-27

**Authors:** Sukanya Unson, Tung-Cheng Chang, Yung-Ning Yang, Shwu-Huey Wang, Chi-Hung Huang, Dana R. Crawford, Haw-Ming Huang, Zi-Lin Li, Hung-Yun Lin, Jacqueline Whang-Peng, Kuan Wang, Paul J. Davis, Wen-Shan Li

**Affiliations:** 1Graduate Institute of Cancer Molecular Biology and Drug Discovery, College of Medical Science and Technology, Taipei Medical University and Academia Sinica, Taipei 11031, Taiwan; d621107005@tmu.edu.tw; 2Division of Colorectal Surgery, Department of Surgery, Taipei Medical University Shuang Ho Hospital, Taipei 11031, Taiwan; 09432@s.tmu.edu.tw; 3Division of Colorectal Surgery, Department of Surgery, School of Medicine, College of Medicine, Taipei Medical University, Taipei 11031, Taiwan; 4School of Medicine, I-Shou University and Department of Pediatrics, E-DA Hospital, Kaohsiung 82445, Taiwan; ed106132@edah.org.tw; 5Core Facility Center, Department of Research Development, Taipei Medical University, Taipei 11031, Taiwan; shwu@tmu.edu.tw; 6Department of Biochemistry and Molecular Cell Biology, College of Medicine, Taipei Medical University, Taipei 11031, Taiwan; 7Division of Cardiology, Department of Internal Medicine, Cathay General Hospital, Taipei 10630, Taiwan; hchbox@cgh.org.tw; 8Department of Immunology and Microbial Disease, Albany Medical College, Albany, NY 12208, USA; crawfod@amc.edu; 9School of Dentistry, College of Oral Medicine, Taipei Medical University, Taipei 11031, Taiwan; hhm@tmu.edu.tw; 10Graduate Institute of Nanomedicine and Medical Engineering, College of Medical Engineering, Taipei Medical University, Taipei 11031, Taiwan; lizilin919@tmu.edu.tw (Z.-L.L.); wangk@tmu.edu.tw (K.W.); 11Graduate Institute of Cancer Molecular Biology and Drug Discovery, College of Medical Science and Technology, Taipei Medical University, Taipei 11031, Taiwan; 12TMU Research Center of Cancer Translational Medicine, Taipei Medical University, Taipei 11031, Taiwan; 13Traditional Herbal Medicine Research Center of Taipei Medical University Hospital, Taipei Medical University, Taipei 11031, Taiwan; 14Cancer Center, Wan Fang Hospital, Taipei Medical University, Taipei 11031, Taiwan; 15Pharmaceutical Research Institute, Albany College of Pharmacy and Health Sciences, Albany, NY 12208, USA; pdavis.ordwayst@gmail.com; 16Department of Medicine, Albany Medical College, Albany, NY 12208, USA; 17Laboratory of Chemical Biology and Medicinal Chemistry, Institute of Chemistry, Academia Sinica, Taipei 10617, Taiwan; wenshan@gate.sinica.edu.tw; 18Doctoral Degree Program in Marine Biotechnology, National Sun Yat-Sen University, Kaohsiung 80424, Taiwan

**Keywords:** heteronemin, tetrac, colorectal cancer, *KRAS*, EGFR signaling

## Abstract

Overexpressed EGFR and mutant *K-Ras* play vital roles in therapeutic resistance in colorectal cancer patients. To search for an effective therapeutic protocol is an urgent task. A secondary metabolite in the sponge *Hippospongia* sp., Heteronemin, has been shown to induce anti-proliferation in several types of cancers. A thyroxine-deaminated analogue, tetrac, binds to integrin αvβ3 to induce anti-proliferation in different cancers. Heteronemin- and in combination with tetrac-induced antiproliferative effects were evaluated. Tetrac enhanced heteronemin-induced anti-proliferation in HT-29 cells (*KRAS* WT CRC) and HCT-116 cells (*KRAS* MT CRC). Heteronemin and tetrac arrested cell cycle in different phases. Combined treatment increased the cell accumulation in sub-G1 and S phases. The combined treatment also induced the inactivation of EGFR signaling and downregulated the phosphorylated ERK1/2 protein in both cell lines. Heteronemin and the combination showed the downregulation of the phosphorylated and total PI3K protein in HT-29 cells (*KRAS* WT CRC). Results by NanoString technology and RT-qPCR revealed that heteronemin and combined treatment suppressed the expression of *EGFR* and downstream genes in HCT-116 cells (*KRAS* MT CRC). Heteronemin or combined treatment downregulated genes associated with cancer progression and decreased cell motility. Heteronemin or the combined treatment suppressed *PD-L1* expression in both cancer cell lines. However, only tetrac and the combined treatment inhibited PD-L1 protein accumulation in HT-29 cells (*KRAS* WT CRC) and HCT-116 cells (*KRAS* MT CRC), respectively. In summary, heteronemin induced anti-proliferation in colorectal cancer cells by blocking the EGFR-dependent signal transduction pathway. The combined treatment further enhanced the anti-proliferative effect via PD-L1 suppression. It can be an alternative strategy to suppress mutant *KRAS* resistance for anti-EGFR therapy.

## 1. Introduction

Colorectal cancer (CRC) is the third most widespread cancer globally [1]. *KRAS* mutations are the most common canonical gain of function mutation in CRC. Oncogenic *KRAS* mutations are evident in 30~40% of CRC patients [2]. *KRAS* is a major oncogene that encodes a small guanosine triphosphatase (GTPase) that binds to the protein and functions to activate signal transduction cascades from cell membranes to the nuclei. Wild-type (WT) KRAS proteins are activated by binding to guanosine triphosphate (GTP), which induces activation signals from the epidermal growth factor (EGF) receptor (EGFR) to downstream effector pathways of phosphatidylinositol 3-kinase (PI3K)/Akt and mitogen-activated protein kinase (MAPK) kinase (MEK)/extracellular signal-regulated kinase (ERK) [3,4].

The overexpression of EGFR accounts for 60~80% of CRC patients, and the higher activity of EGFR was inversely associated with the survival rate [5]. In CRC pathogenesis, the EGFR signaling pathway is related to promoting cancer cell proliferation, migration, apoptosis inhibition, angiogenesis [6], and immune evasion [7]. The anti-EGFR monoclonal antibodies, cetuximab, and panitumumab are current targeted therapies for patients diagnosed with *KRAS* WT metastatic CRC [8]. Anti-EGFR monoclonal antibodies are ineffective in CRC patients with a *KRAS* mutation due to cancer’s genetic heterogeneity. The *KRAS* status is a major criterion when considering the use of targeted therapies. Moreover, the activation of the EGFR signaling pathway is related to the host antitumor immunity through programmed death ligand-1 (PD-L1) upregulation [9]. PD-L1 expression more often occurs in metastatic CRC, and its expression in primary CRC might not indicate that cancer cells have spread to distant locations in the body [10]. Therefore, finding new therapeutic strategies that show anticancer efficacy and overcome anti-EGFR therapeutic resistance is an important challenge in *KRAS* mutant (MT) CRC.

Heteronemin, a marine sesterterpene derived from the sponge *Hippospongia* sp., is a natural marine product that has various pharmacological effects such as anticancer, anti-inflammatory, antioxidant activities, and apoptotic effects [11,12,13,14]. In preclinical studies, heteronemin was a potent inhibitor of tumors of several cancer types, including oral cancer [14], prostate cancer [15], kidney cancer [11], lung cancer [16], myeloid leukemia [17], cholangiocarcinomas [18], and breast cancer [19]. In human renal carcinoma cells, heteronemin induces antiproliferative activity by suppressing Akt phosphorylation, and ERK1/2 expression, and sequentially inducing cell apoptosis [11]. Furthermore, it was shown to improve the efficacy of cytarabine by inhibiting the farnesylation of the Ras protein. Sequentially, it inhibited the MAPK pathway by inactivating ERK1/2 and c-Myc proteins to prevent tumor cell proliferation [17]. Heteronemin has been shown to bind cell surface integrin αvβ3 receptor [20] and inhibit downstream ERK1/2 and STAT3 in breast cancer [19]. 

Tetraiodothyroacetic acid (tetrac) is a deaminated analog of the thyroid hormone (T_4_) that blocks the proangiogenic activity of the thyroid hormone via competing with the cell surface integrin αvβ3 receptor [21]. Tetrac inhibits thyroid hormone-induced angiogenesis and proliferation [21]. Tetrac suppresses the T_4_-induced activation of ERK1/2 and STAT3 in lung cancer [22]. Additionally, tetrac restricted tumor cell proliferation in a breast cancer xenograft model [23]. Tetrac induces an antiproliferative effect by suppressing ERK1/2 activation in colorectal cancer cell lines with different *KRAS* statuses [24]. However, the mechanisms of anti-proliferation of heteronemin and tetrac in CRC are still unknown. The purpose of this study is to investigate whether tetrac enhanced the antitumor effect of heteronemin in CRC with different *KRAS* statuses by blocking EGFR signaling and its downstream signal transduction pathways. In addition, the combined treatment inactivated integrin αvβ3-dependent signaling. The combined treatment modulated gene expressions in HCT-116 cells (*KRAS* MT CRC) shown by nanostring technology. Sequentially, heteronemin, tetrac, and the combined treatment induced anti-proliferation in both HT-29 cells (*KRAS* WT CRC) and HCT-116 cells (*KRAS* MT CRC).

## 2. Results

### 2.1. The Combined Treatment of Heteronemin and Tetrac Inhibits Cell Proliferation in Human CRC Cells with Different KRAS Statuses

Heteronemin has been shown to induce antiproliferative properties against various types of cancers [20]. The HT-29 cells (*KRAS* WT CRC) and HCT-116 cells (*KRAS* MT CRC) express different gene mutation statuses [25]. Initially, we assessed the inhibitory effects of heteronemin in both CRC cell lines using a CyQUANT^®^ cell proliferation assay after 24 and 72 h of treatment. The results show that treatment with heteronemin effectively inhibited cell proliferation in both CRC cell lines in a concentration- and time-dependent manner (Figure 1A,B). Moreover, HCT-116 cells (*KRAS* MT CRC) were more sensitive to heteronemin than HT-29 cells (*KRAS* WT CRC). The 50% inhibitory concentration (IC_50_) values of heteronemin were 2.4 and 0.8 µM for HT-29 cells (*KRAS* WT CRC) and 1.2 and 0.4 µM for HCT-116 cells (*KRAS* MT CRC) at 24 and 72 h, respectively.

Tetrac inhibits tumor growth [24]. It also suppresses the proangiogenic actions of vascular endothelial growth factor (VEGF) [21]. Tetrac also inhibits tumor growth in the HCT-116 (*KRAS* MT CRC) mouse xenograft model [26]. Tetrac at 10^−7^ M suppresses the ERK1/2 phosphorylation in HT-29 cells (*KRAS* WT CRC) and HCT-116 cells (*KRAS* MT CRC) [24]. We further investigated the inhibitory effects of heteronemin, tetrac, and their combination in both CRC cell lines. Cells were treated with tetrac at 10^−6^ and 10^−7^ M, or combined with heteronemin at 0.4 or 0.8 µM (HT-29 cells; *KRAS* WT CRC) and 0.2 or 0.4 µM (HCT-116 cells; *KRAS* MT CRC) for 72 h (Figure 1C,D). The results demonstrate that heteronemin and tetrac significantly inhibited cell proliferation, and tetrac further improved the inhibitory effect of heteronemin in both cell lines. Moreover, the combined effect of the two compounds produced a greater reduction in cell proliferation compared to heteronemin alone. A total of 0.8 µM of heteronemin in HT-29 cells (*KRAS* WT CRC) or 0.4 µM of heteronemin in HCT-116 cells (*KRAS* MT CRC) and 10^−7^ M of tetrac were further used to identify whether tetrac enhanced the antitumor effect of heteronemin in human CRC cells with different *KRAS* statuses.

### 2.2. The Combination of Heteronemin and Tetrac Alters Cell Cycle Arrest at the Sub-G_1_ and S Phases in Human CRC Cells with Different KRAS Statuses

The suppression of tumor cell growth is commonly implicated in cell cycle arrest. To evaluate the combined treatment with heteronemin and tetrac, we further determined the effects of their combination in HT-29 cells (*KRAS* WT CRC) and HCT-116 cells (*KRAS* MT CRC) using flow cytometry. As shown in Figure 2, the cell cycle histograms revealed that heteronemin caused a significant accumulation of the cell population in the sub-G_1_ phase (apoptotic cells) in both cell lines and the S phase in HT-29 cells (*KRAS* WT CRC). Tetrac at 10^−7^ M for 24 h led to accumulation in the S phase compared to the control. Interestingly, tetrac treatment significantly increased the S-phase cell population from 8.7% (control) to 14.0% in HT-29 cells (*KRAS* WT CRC) and from 13.7% (control) to 16.8% in HCT-116 cells (*KRAS* MT CRC) (Figure 2A,B). The data suggest that S-phase arrest contributes to the antiproliferative effect of tetrac in both cell lines.

Furthermore, we investigated whether tetrac could enhance the therapeutic effect of heteronemin. The cell-cycle profiles of these CRC cell lines showed similar trends in the sub-G_1_ and S phases after combined treatment compared to their respective controls. The S phase population in HT-29 cells (*KRAS* WT CRC) and HCT-116 cells (*KRAS* MT CRC), respectively, rose from 12.2% to 15.1% and from 10.6% to 17.8% compared to treatment with heteronemin alone (Figure 2A,B). 

### 2.3. The Combined Treatment of Heteronemin and Tetrac Suppresses the Phosphorylation of EGFR in Human CRC Cells with Different KRAS Statuses

The RAS protein mediates signals from growth factor receptors (such as EGFR) to downstream effector pathways, such as PI3K and ERK1/2. The EGFR pathway played a key role in the pathogenesis, cell proliferation, and immune evasion of cancer [27]. We first assessed the EGFR protein expression levels after treatment with heteronemin, tetrac, or heteronemin in combination with tetrac for 24 h. As illustrated in Figure 3A,B, the phosphorylation of EGFR was significantly downregulated in both cell lines treated with tetrac or combined treatment compared with the control group. In *KRAS* MT cells, tetrac or combined treatment caused dramatic reductions in the total EGFR levels (Figure 3B). Interestingly, the combined treatment also led to a strong reduction in total and phosphorylated EGFR protein compared to heteronemin alone in HCT-116 cells (*KRAS* MT CRC). These results demonstrated that combination treatment was more efficient in terms of suppressing EGFR activation and EGFR expression in both CRC cell lines. 

### 2.4. The Combined Treatment of Heteronemin and Tetrac Downregulates ERK1/2 Phosphorylation and ERK1/2 Protein Levels in Human CRC Cells with Different KRAS Statuses

The EGFR regulates two critical components, PI3K/AKT and MAPK/ERK1/2, of intracellular signaling, which have essential functions in cell proliferation. To understand which molecular mechanisms of EGFR downstream effectors were affected by heteronemin combined with tetrac, and it may link to PD-L1 expression in CRC cells with different *KRAS* statuses. We, therefore, analyzed phosphorylated and total protein levels of PI3K and ERK1/2.

The results show that the phosphorylated and total protein expression levels of PI3K were significantly downregulated in HT-29 cells (*KRAS* WT CRC) with combination treatment compared to the control for 24 h (Figure 4A), whereas there were unaffected in HCT-116 cells (*KRAS* MT CRC; Figure 4B). Surprisingly, the expression levels of the phosphorylated and total ERK protein were dramatically suppressed after combined treatments compared to the control in HCT-116 cells. Interestingly, single-agent heteronemin or combined with tetrac showed a significant reduction in phosphorylated ERK1/2 in HT-29 cells (*KRAS* WT CRC). Collectively, these results suggest that heteronemin combined with tetrac inhibited PI3K and ERK1/2 activation in HT-29 cells (*KRAS* WT CRC) and ERK1/2 in HCT-116 cells (*KRAS* MT CRC), which correlated with the suppression of EGFR. 

### 2.5. Heteronemin, Tetrac, and Their Combination Regulate Gene Expression in KRAS Mutant Cells

*K-Ras* play critical roles in therapeutic resistance in CRC patients. To investigate gene expression profiling involved in the anti-proliferative effect induced by tetrac, heteronemin, and the combined treatment in HCT-116 cells (*KRAS* MT CRC), NanoString nCounter was used for gene expression analyses. It provides a robust, highly reproducible, and sensitive method. In addition, it has the capacity to evaluate the direct digital readout of up to 800 gene targets from a sample in a single sample. It uses fluorescent barcode-labeled hybridization probes without reverse transcription. The results of the gene expression of HCT-116 cells (*KRAS* MT CRC) by NanoString gene expression analyses are displayed as a heatmap of the directed global significance score that was involved in cancer progression in HCT-116 cells (*KRAS* MT CRC). Pathway scores were identified by nSolverTM software. Most signal pathways were upregulated by heteronemin or combination treatment. Heteronemin combined with tetrac appeared to reduce the expression level of signal transduction pathways involved in EGFR signaling, which includes cell proliferation, angiogenesis, and epithelial–mesenchymal transition (EMT) in tumor metastasis compared to heteronemin alone. In contrast, cell motility showed an inverse expression pattern of all signal transduction pathways that drive cancer progression (Figure 5A,B). 

The activated EGFR signal transduction pathway induces cellular motility [28]. In addition, EMT plays a vital role in tumor metastasis [29,30]. Thus, cell motility is associated with EMT in cancer invasion and metastasis [31]. It was of interest to investigate if heteronemin, tetrac, and the combined treatment modulated gene sets that are involved in cell motility, i.e., *EGFR* and signal transduction and activator of transcription 3 (*STAT3*), as two of the gene lists (Appendix A). These observations indicated that heteronemin and combined treatment induced the downregulation of *EGFR* and *STAT3* mRNA expressions.

EGFR activation is related to cancer progression. The pathway scores of cell proliferation, cellular growth factor, cellular differentiation, choline cancer metabolism, and carbon cancer metabolism were shown to have functional statuses with the EGFR. Therefore, heteronemin alone and combined with tetrac induced upregulated correlations for all pathway scores of the EGFR in HCT-116 cells (*KRAS* MT CRC). However, only cell motility showed an inverse correlation among pathway scores of the EGFR after heteronemin or combination treatment (Figure 6).

Additionally, the expressions of a set of genes associated with the cell cycle were investigated. Among differentially expressed genes (DEGs) in the heatmap, those genes were related to the effect of heteronemin or tetrac modulated cell cycle progression in HCT-116 cells (*KRAS* MT CRC; Appendix A). 

We also examined changes in the expressions of genes involved in antiproliferation induced by heteronemin, tetrac, and their combination in HCT-116 cells (*KRAS* MT CRC) for 24 h. The mRNA levels of 771 genes were analyzed by a Nanospring^®^ nCounter PanCancer Progression Panel (NanoString Technologies, Seattle, WA, USA). Table 1 shows the significant modulation of genes after treatment with heteronemin, tetrac, or their combination. Additionally, the expressions of genes and growth factor pathways associated with CRC progression, such as *EGFR, TGFB1,* transforming growth factor-β receptor 2 (*TGFBR2*), and tumor protein 53 (*TP53*), were significantly reduced by combined treatment (Table 2). Although *KRAS* expression was not statistically significantly reduced, it showed decreased expressions after different treatments. These studies illustrated that *EGFR* might be an important target of heteronemin and tetrac. 

### 2.6. The Combined Treatment of Heteronemin and Tetrac Downregulated EGF and EGFR mRNA Expressions in Human CRC Cells with Different KRAS Statuses

We next assessed the mRNA expressions of the most important EGFR pathways, including *EGF, EGFR, ERK1/2,* and *PI3K*. When *KRAS* WT cells were treated with heteronemin alone, this significantly reduced levels of *EGF* and *EGFR* mRNA compared to the control (Figure 7A,C). Single-agent tetrac, with or without heteronemin, caused significant decreases in *EGF* and *EGFR mRNA* levels in both cell lines compared to the control (Figure 7A–D). In addition, *EGF* and *EGFR* levels dramatically decreased after being treated with the combination compared to single-agent heteronemin in HT-29 cells (*KRAS* WT CRC) and HCT-116 cells (*KRAS* MT CRC), respectively (Figure 7A,D). *ERK1/2* was found to be decreased after tetrac in HT-29 cells (*KRAS* WT CRC). Although *PI3K* mRNA levels did not exhibit any statistically significant difference, a decreasing trend was shown after being treated with heteronemin or tetrac in both cell lines, and by combined treatment in HCT-116 cells (*KRAS* MT CRC).

### 2.7. The Combination of Heteronemin and Tetrac Shows Different Regulations of Cell Cycle Regulators in Human CRC Cells with Different KRAS Statuses

*Cyclin D1* and *PCNA* are specific genes, respectively, active in the G_1_ and S phases of the cell cycle. These genes are downstream cell cycle regulation targets of c-Myc [32]. We determined the mRNA and protein levels of these genes, which are linked to the inhibition of cell proliferation in HT-29 cells (*KRAS* WT CRC) and HCT-116 cells (*KRAS* MT CRC). Treatment with heteronemin, tetrac, or their combination reduced significantly mRNA levels of *cyclin D1* in both cell lines (Figure 8A). Heteronemin alone significantly inhibited *PCNA* expression in both cell lines, and the combination of heteronemin and tetrac also showed a significant reduction in *PCNA* mRNA in HCT-116 cells (*KRAS* MT CRC). In addition to RNA expression, levels of the cyclin D1 and PCNA proteins were not significantly altered, but combination treatment produced reductions in their protein expressions in HCT-116 cells (*KRAS* MT CRC). The amount of *c-Myc* mRNA was significantly downregulated after treatment with tetrac or its combination with heteronemin in both cell lines compared to the control (Figure 8A). Consequently, Western blot analysis also showed similar results in terms of c-Myc protein expression after tetrac or combined treatment in HT-29 cells (*KRAS* WT CRC; Figure 8B). These results indicate that heteronemin combined with tetrac blocked cell cycle progression through the inactivation of c-Myc in HT-29 cells (*KRAS* WT CRC). 

### 2.8. The Combined Treatment of Heteronemin and Tetrac Suppresses the Expression of *PD-L1* in Human CRC Cells with Different KRAS Statuses

The activation of the EGFR signaling pathway is related to the host antitumor immunity through PD-L1 upregulation [9]. PD-L1 activation is related to the modulation of host immune responses [33]. Modulating EGFR activation also regulates *PD-L1* expression [9]. We wondered if the inhibitory effect of heteronemin on PD-L1 expression was enhanced by tetrac link to EGFR signaling in CRC cells. We measured the levels of PD-L1 mRNA and protein for 24 h. Heteronemin or combined with tetrac significantly decreased *PD-L1* mRNA levels in both cell lines. Western blot analyses revealed that combined treatment significantly suppressed the PD-L1 protein level in HCT-116 cells (*KRAS* MT CRC), consistent with the results of the *PD-L1* mRNA level (Figure 9B,D). Similarly, there was a significant decrease in PD-L1 mRNA and protein levels after tetrac treatment in HT-29 cells (*KRAS* WT CRC; Figure 9A,C). These results demonstrate that combination treatment showed a link between EGFR/ERK1/2 signaling and PD-L1 in HCT-116 cells (*KRAS* MT CRC). 

## 3. Discussion

Anti-EGFR monoclonal antibodies, including cetuximab and panitumumab, are used as targeted therapy for CRC treatment. Additionally, targeted drugs interfere with or inhibit specific proteins that are involved in cancer cell growth. However, these drugs are ineffective in patients with *KRAS* or *BRAF* mutations. Their efficacy depends on the genetic profile of CRC patients (WT or MT *KRAS* and *BRAF*) [34]. It was reported that heteronemin had an anticancer effect on several types of cancers [11,14,15,16,17,18]. Furthermore, targeting integrin αvβ3 by tetrac and its derivatives displays anticancer effects, such as abrogating angiogenesis and inhibiting tumor growth [11,14,15,16,17,18]. Therefore, the current study aimed to evaluate whether tetrac could improve the antitumor effect of heteronemin in CRC with different *KRAS* statuses. Our data demonstrated that tetrac enhanced the antitumor effect of heteronemin in HT-29 cells (*KRAS* WT CRC) and HCT-116 cells (*KRAS* MT CRC; Figure 1A,B).

The inhibition of tumor cell growth is usually mediated through cell cycle arrest. Previously, 2.5 and 5 µM heteronemin were shown to alter the distribution of apoptotic cells at the sub-G_1_ stage in HeLa cells [35]. Similar to that study, 0.4 and 0.8 µM heteronemin raised the percentage of the cell population at the sub-G_1_ phase in CRC cell lines (Figure 2A,B). We recently demonstrated that different concentrations of heteronemin altered different cell cycle stages, and the accumulation of cells in the G_1_ and G_2_/M stages increased with heteronemin combined with tetrac in OEC-M1 and SCC-25 cells, respectively [14]. In our study, combined treatment increased HT-29 (*KRAS* WT CRC) and HCT-116 (*KRAS* MT CRC) cell accumulation at the sub-G_1_ and S phases of the cell cycle compared to the control (Figure 2A,B). Although there were similar trends of cell cycle distributions between CRC cells with different *KRAS* statuses, this combination reduced the accumulation of cells in the sub-G_1_ phase compared to heteronemin alone in HT-29 cells (*KRAS* WT CRC). 

As for EGFR-mediated signaling in the mutant *KRA**S*, the EGFR/RAS/ERK and EGFR/RAS/PI3K signaling pathways contribute to cell proliferation and angiogenesis [36]. A previous study reported that heteronemin enhanced the antiproliferative activity of cytarabine by blocking farnesylation of the Ras protein leading to the downregulation of MAPK, activator protein (AP)-1, nuclear factor (NF)-κB, and c-Myc activation [17]. Tetrac induced the upregulation of gene expressions of both the antiangiogenic thrombospondin 1 (*THBS1*) and antiapoptotic *XIAP* in human breast cancer cells [37]. Thus, our data show that heteronemin, tetrac, and their combination significantly decreased expressions of *EGF* and *EGFR* mRNA in both cell lines with different *KRAS* statuses, except for heteronemin alone in HCT-116 cells (*KRAS* MT CRC; Figure 7A–D). In contrast, neither *ERK1/2* nor *PI3K* significantly changed following treatment with single or combined agents. As to the detection of protein levels, we found that phosphorylated EGFR expression was significantly downregulated in both cell lines after 24 h of incubation with tetrac or its combination with heteronemin (Figure 3A), while total EGFR protein was significantly downregulated after combined treatment in HCT-116 cells (*KRAS* MT CRC; Figure 3B). The combined treatment suppressed PI3K activation in HT-29 cells (*KRAS* WT CRC). In hepatocellular carcinoma cell lines, antiproliferative activity was induced by heteronemin, which downregulated ERK/MAPK expressions and also promoted the formation of reactive oxygen species (ROS) [38]. We have recently shown that heteronemin suppressed ERK1/2 activation in breast cancer cells [19], similar to this study in HT-29 cells (*KRAS* WT CRC; Figure 4A). Additionally, this combination also reduced the ERK1/2 protein expressions in both CRC cell lines (Figure 4A,B).

NanoString nCounter technology was used to analyze gene expression profiling involved in cancer progression in HCT-116 cells (*KRAS* MT CRC). Our previous research showed that heteronemin modulated the expression levels of genes associated with signal transduction pathways in cancer progression, including cell motility and angiogenesis in cholangiocarcinoma cell lines (HuccT1 and SSP-25 cells) by NanoString technology [18]. Similar to that study, the downregulation of cell motility and *EGFR* gene expressions were reduced by heteronemin and combined with tetrac in HCT-116 cells (*KRAS* MT CRC; Figure 5 and Appendix A, Table 2). EGFR signaling pathways have been shown to regulate a series of important events, such as cell proliferation, angiogenesis, EMT, and tumor metastasis in CRC [6,30]. This study illustrated that tetrac improved the effect of heteronemin through reduction in those signaling pathway expressions compared to heteronemin (Figure 5A,B). Although the expressions of CCD1, PCNA, and c-Myc proteins did not show significant downregulation after treatments in HCT-116 cells (*KRAS* MT CRC), they might be involved in alterations of the list of genes (Appendix A).

It was observed that PD-L1 expression was more frequent in *RAS* or *TP53* WT CRC than in *RAS*- or *TP53*-mutated CRC [39]. Moreover, upregulated PD-L1 was more highly expressed on the cell surface of *KRAS* mutant CRC. [40]. The thyroid hormone induces PD-L1 expression via ERK1/2 activation [41]. The nanoparticulate analog nano-diamino-tetrac (NDAT) inhibited PD-L1 expression in HCT-116 (*KRAS* MT CRC) xenograft model [42]. Our results show that the downregulation of PD-L1 expression was modulated by heteronemin combined with tetrac, and its expression was also linked to blockage of the EGFR/ERK1/2 signaling cascade in HCT-116 cells (*KRAS* MT CRC; Figure 9B). Conversely, the modulation of PD-L1 protein expression was significantly changed after tetrac alone in HT-29 cells (*KRAS* WT CRC; Figure 9A). Likewise, HCT-116 cells (*KRAS* MT CRC) were suppressed via the downregulation of the EGFR and ERK1/2 proteins. Combined treatment showed the impact of *KRAS* mutation status on the EGFR signaling pathway and antitumor immune response. Some gene expressions were not consistent with mRNA and protein expressions, including *PI3K* and *ERK1/2.* The differences in correlation between the percentage of mRNA and protein expression are caused by transcription and translation processes. 

## 4. Materials and Methods

### 4.1. Chemicals

Heteronemin and tetraiodothyroacetic acid (purity >98%) were obtained from Sigma-Aldrich (St. Louis, MO, USA). Other chemicals used were of analytical grade or higher.

### 4.2. Cell Culture

Human CRC HT-29 cells (*KRAS* WT CRC) and HCT-116 cells (*KRAS* MT CRC) were purchased from American Type Culture Collection (ATCC^®^, Manassas, VA, USA) by the Bioresource Collection and Research Center (BCRC, Hsinchu, Taiwan). Cells were maintained in RPMI 1640 medium (Gibco^®^, Invitrogen, New York, NY, USA), supplemented with 10% fetal bovine serum, 100 U/mL penicillin, and 100 µg/mL streptomycin, under incubation conditions of 5% CO_2_ at 37 °C. Cells were placed in a 0.25% hormone-depleted serum-supplemented medium for 24 h of starvation before treatment.

### 4.3. Cell Proliferation Assay

Cell viability was evaluated by a CyQUANT^®^ cell proliferation assay in which HT-29 cells (*KRAS* WT CRC) and HCT-116 cells (*KRAS* MT CRC) were seeded in six-well plates at a density of 10^5^ cells/well. Cells were incubated overnight and then synchronized by 24 h of starvation. Cells were treated with 0.25% serum-stripped medium (vehicle control) or different concentrations of heteronemin, tetrac, or their combination for 24 and 72 h. After treatment, 50 µL of each sample was transferred in triplicate into 96-well plates in three independent experiments. Cells were supplemented with 100 µL of CyQUANT^®^ NF reagent for 1 h at 37 °C. Thereafter, plates were detected using a microplate reader (Varioskan, ThermoFisher Scientific, Waltham, MA, USA), and the absorbance was measured with excitation at 485 nm and emission at 530 nm.

### 4.4. Cell Cycle Analysis 

Cells were plated in six-well plates at a density of 10^5^ cells/well followed by treatment with heteronemin, tetrac, or their combination for 24 h. Cells were harvested by trypsinization, fixed with 70% ethanol at −20 °C, washed with phosphate-buffered saline (PBS), and treated with 1 mL of Triton X-100 containing RNase A for 1 h. A propidium iodide (PI) solution (0.05 mg/mL) was used for cell staining in the dark for 30 min. Cells were analyzed using an Invitrogen Attune™ NxT Acoustic Focusing Cytometer (ThermoFisher Scientific, Waltham, MA, USA). The particular phase of the cell cycle with DNA content in the sub-G_1_, G_1_, S, and G_2_/M phases was estimated by Attune NxT Flow Cytometer software (vers. 4.2).

### 4.5. Western Blot Analysis

After the treatment period for each experiment, cells were lysed with cold RIPA lysis buffer, containing protease inhibitors (Sigma, St. Louis, MO, USA). Equal concentrations of protein samples (30 µg) were run on sodium dodecylsulfate polyacrylamide gel electrophoresis (SDS-PAGE), and transferred to polyvinylidene fluoride (PVDF) membranes, and blocked with 5% non-fat milk for 1 h. Thereafter, membranes were incubated with primary antibodies overnight at 4 °C. The following primary antibodies were used: rabbit anti-phospho-EGFR (1:1000), rabbit anti-cyclin D (1:1000), rabbit anti-PCNA (1:3000), mouse anti-GAPDH (1:10,000; GeneTex International, Hsinchu City, Taiwan), rabbit anti-EGFR (1:1000), rabbit anti-phospho-ERK1/2 (1:1000), rabbit anti-ERK1/2 (1:1000), rabbit anti-*PD-L1*(1:1000), rabbit anti-c-Myc (1:1000; Cell Signaling Technology, Beverly, MA, USA), rabbit anti-phospho-PI3K (1:2000; Abcam, Cambridge, MA, USA), and mouse anti-PI3K (1:1000; BD Bioscience, Franklin Lakes, NJ, USA). Membranes were washed in washing buffer and then incubated with a secondary antibody for 1 h. Protein bands were detected using images of Western blots visualized and recorded with an Amersham Imager 600 (GE Healthcare, Chicago, IL, USA). Blots were quantified using ImageJ software (National Institutes of Health, Bethesda, MD, USA).

### 4.6. NanoString Gene Expression Analysis 

HCT-116 cells (*KRAS* MT CRC) were seeded in six-well plates overnight at a density of 10^5^ cells/well. Cells were then treated with heteronemin, tetrac, or their combination for 24 h. Total RNA was isolated using the TRIzol reagent. For quality assurance, RNA samples with an RNA integrity number (RIN) of >5 were used for the NanoString^®^ analysis. An nCounter PanCancer Progression Panel was used. Technical and biological normalization of the raw counts of each gene was performed using nSolver Software vers. 4.0. For technical normalization, a positive control factor was calculated for each sample. All normalization steps were performed using a Nanostring nCounter software analysis. 

### 4.7. Reverse-Transcription Quantitative Polymerase Chain Reaction (RT-qPCR)

Total RNA was isolated from cultured cells, and an RNAspin Mini RNA Isolation Kit (Macheren-NAGEL, Duren, Germany) was used to remove genomic DNA. RevertAid H Minus First Strand cDNA Synthesis Kit (Thermo Scientific, Vilnius, Lithuania) was used for the synthesis of complementary (c)DNA from total RNA templates, according to the manufacturer’s instruction. cDNA was used as the template for real-time PCRs and analyses. Real-time PCRs were carried out using a QuantiNova™ SYBR^®^ Green PCR Kit (Qiagen, Valencia, CA, USA). The samples had a final volume of 20 µL containing 1X SYBR Green Supermix and 0.7 µM for forward and reverse primer of each target gene. cDNA was run on a CFX Connect™ Real-Time PCR Detection System (Bio-Rad Laboratories, Hercules, CA, USA) with the following cycling conditions: 95 °C for 2 min, followed by 40 cycles of 95 °C for 5 min and 60 °C for 10 s. The messenger (m)RNA level of each target gene was normalized to GAPDH [43]. Specific primers are listed in Table 3.

### 4.8. Statistical Analysis

Data were analyzed with IBM^®^ SPSS^®^ statistics vers. 23 (SPSS, Armonk, NY, USA). Data are expressed as the mean ± standard deviation (SD). Data were statistically analyzed with a one-way analysis of variance (ANOVA) followed by post hoc Tukey’s honest significant difference (HSD) test for multiple group comparisons. *p*-values of < 0.05 were considered statistically significant.

## 5. Conclusions

Tetrac effectively improved the antiproliferative activity of heteronemin through the inactivation of the EGFR/ERK1/2 signaling pathway and induction of antitumor immunity via suppression of PD-L1 expression in HCT-116 cells (*KRAS* MT CRC). Moreover, ERK1/2 inactivation in HCT-116 cells (*KRAS* MT CRC) showed crosstalk between EGFR signaling and PD-L1 expressions. In contrast, combination treatment blocked EGFR activation and molecules downstream of the PI3K and ERK1/2 pathways in HT-29 cells (*KRAS* WT CRC). The different *KRAS* statuses had an impact on drug efficacy in particular CRC cell lines. The combination of tetrac and heteronemin might potentially be used as an alternative strategy to increase the effectiveness of acquired resistance to anti-EGFR therapy due to *KRAS* mutations. However, more preclinical and clinical studies need to be performed to determine the safety and effectiveness of this combination.

## Figures and Tables

**Figure 1 marinedrugs-20-00482-f001:**
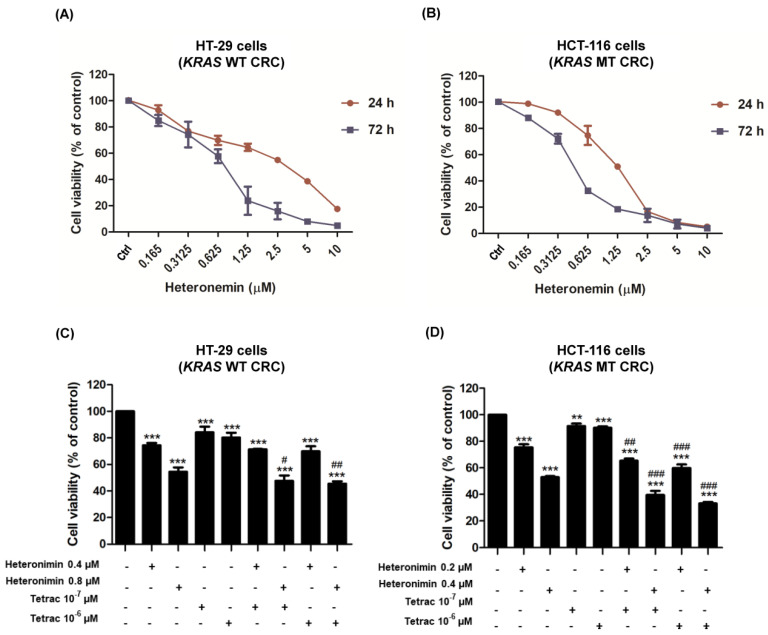
Tetrac enhances heteronemin-induced cytotoxicity in HT-29 cells (*KRAS* WT CRC) and HCT-116 cells (*KRAS* MT CRC). HT-29 cells (**A**) and HCT-116 cells (**B**) were treated with different concentrations of heteronemin for 24 and 72 h. IC_50_ values were calculated using GraphPad Prism V5.0. Cells were treated with heteronemin at IC_50_ concentrations, tetrac, or their combination for 72 h (**C**,**D**). The viabilities of HT-29 and HCT-116 cell lines were evaluated with a CyQUANT^®^ cell proliferation assay. Data are presented as the mean ± SD of two or three independent experiments performed in triplicate. ** *p* < 0.01, and *** *p* < 0.001 compared with untreated control; ^#^
*p* < 0.05, ^##^
*p* < 0.01 and ^###^
*p* < 0.001 compared with heteronemin.

**Figure 2 marinedrugs-20-00482-f002:**
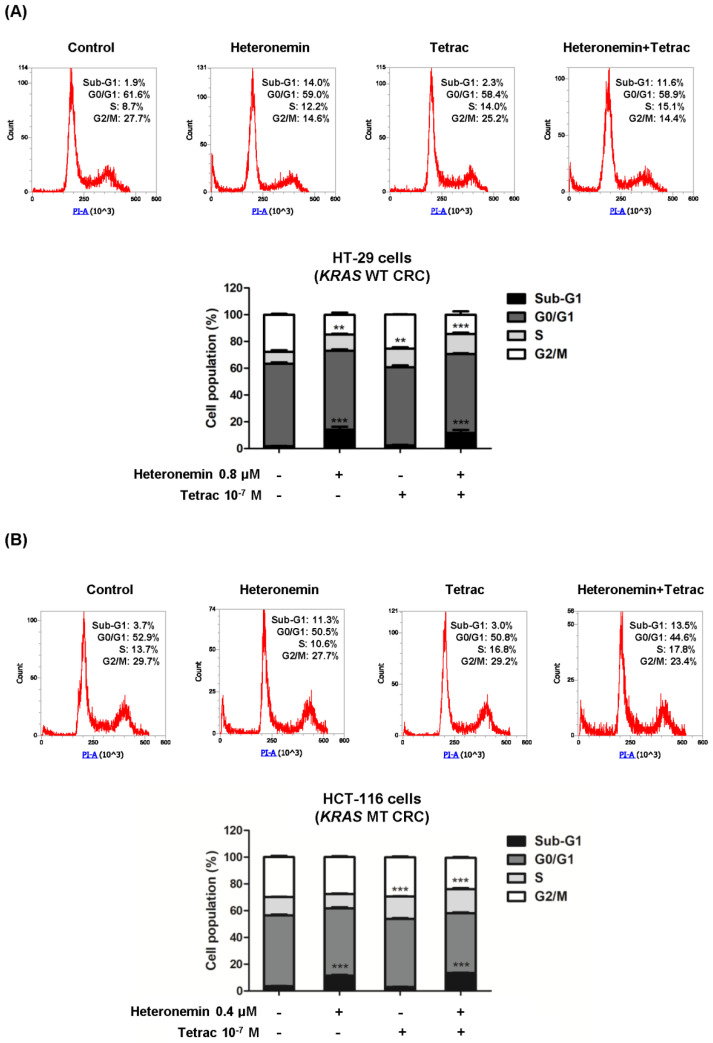
Combined treatment of heteronemin and tetrac alters the cell cycle in HT-29 cells (*KRAS* WT CRC) and HCT-116 cells (*KRAS* MT CRC). HT-29 cells (**A**) and HCT-116 cells (**B**) were treated with heteronemin, tetrac, or their combination for 24 h. Cells were harvested for cell cycle analysis by flow cytometry. Sub-G_1_, apoptosis phase; G_0_/G_1_, pre-DNA synthetic phase, and stationary phase; S, DNA synthesis phase; G_2_/M, post-DNA synthetic phase, and mitotic phase. Data are presented as the mean ± SD (*n* = 3). ** *p* < 0.01 and *** *p* < 0.001 compared with untreated control.

**Figure 3 marinedrugs-20-00482-f003:**
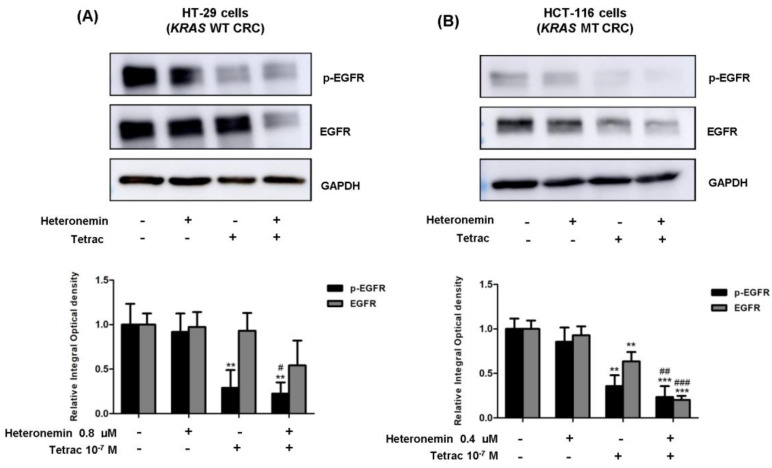
Combined treatment of heteronemin and tetrac alters the expression levels of EGFR. Western blot analyses showed changes in phosphorylated and total protein levels for 24 h in HT-29 cells (*KRAS* WT CRC) (**A**) and HCT-116 cells (*KRAS* MT CRC) (**B**). GAPDH was used as a loading control for protein normalization. Data are presented as the mean ± SD of three independent experiments. ** *p* < 0.01 and *** *p* < 0.001 compared with the untreated control; ^#^
*p* < 0.05, ^##^
*p* < 0.01 and ^###^
*p* < 0.01 compared with heteronemin.

**Figure 4 marinedrugs-20-00482-f004:**
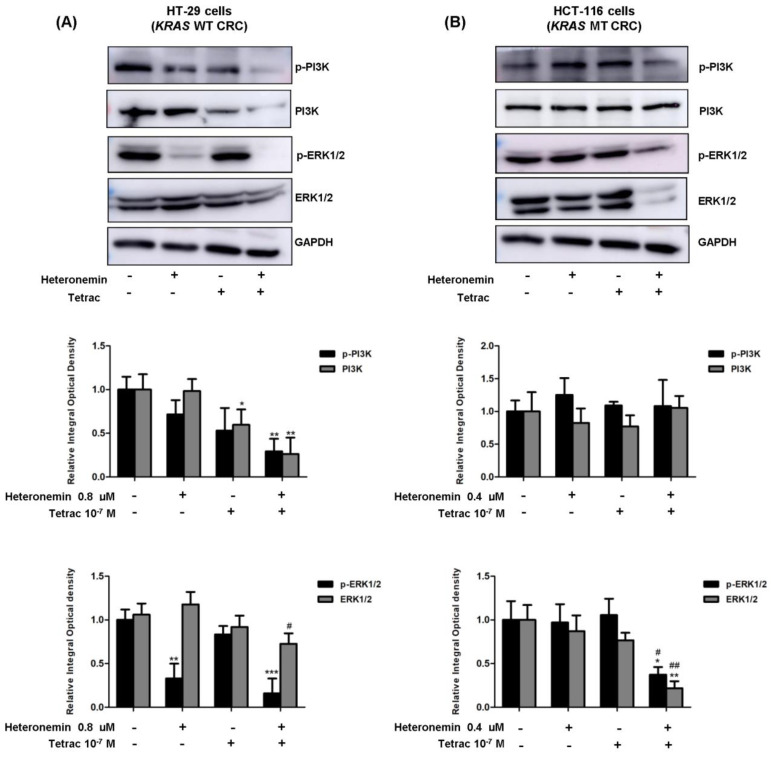
Combined treatment of heteronemin and tetrac modulates protein levels of PI3K and ERK1/2. Western blot analyses showed changes in phosphorylated and total protein levels for 24 h in HT-29 cells (*KRAS* WT CRC) (**A**) and HCT-116 cells (*KRAS* MT CRC) (**B**). GAPDH was used as a loading control for protein normalization. Data are presented as the mean ± SD of three independent experiments. * *p* < 0.05, ** *p* < 0.01 and *** *p* < 0.001 compared with the untreated control; ^#^
*p* < 0.05 and ^##^
*p* < 0.01 compared with heteronemin.

**Figure 5 marinedrugs-20-00482-f005:**
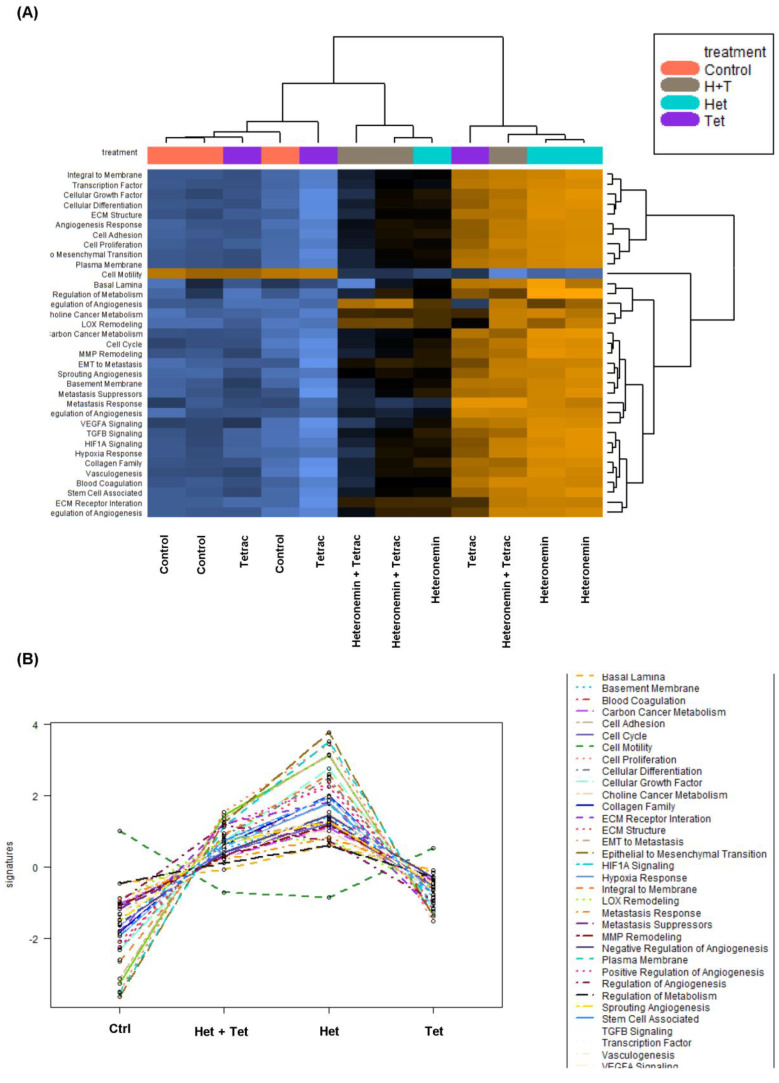
Heatmap of signal transduction pathways that showed significant (*p* < 0.05) upregulation or downregulation. The yellow region of the heatmap indicates upregulation, the blue region indicates downregulation, and the black indicates medial expression (**A**). The differential expression in each gene set represents the treatment compared with the control group. Correlations between signaling pathways in HCT-116 cells (*KRAS* MT CRC) after 24 h of treatment (**B**). The lower panel represents the mean of each treatment in the upper heatmap.

**Figure 6 marinedrugs-20-00482-f006:**
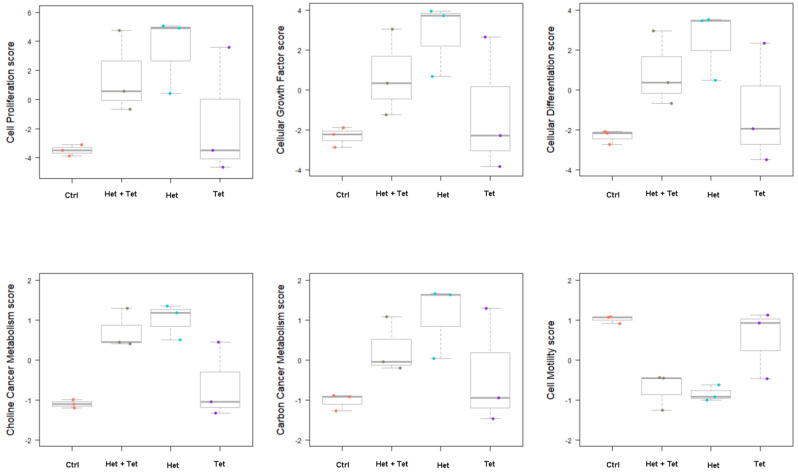
Pathway scores showed functional statuses of EGFR in HCT-116 cells (*KRAS* MT CRC). Pathway scores were analyzed by nSolverTM software for comparisons between the treated and control groups.

**Figure 7 marinedrugs-20-00482-f007:**
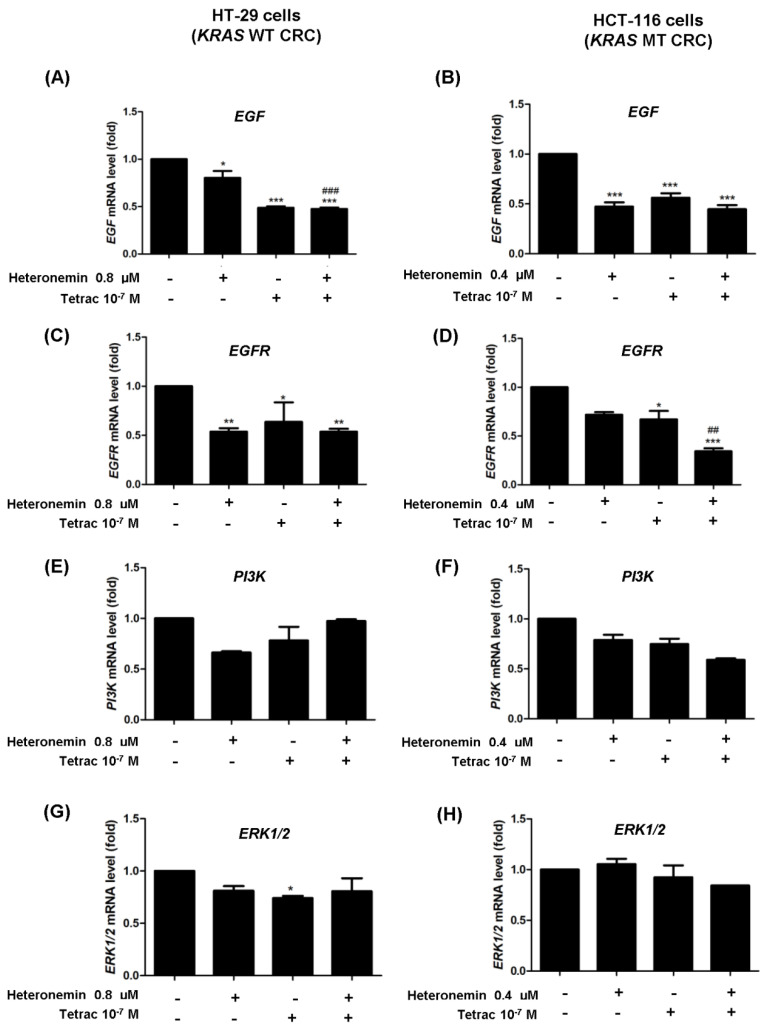
The combined treatment of heteronemin and tetrac regulates gene expressions. The levels of epidermal growth factor (EGF) (**A**,**B**), EGF receptor (EGFR) (**C**,**D**), phosphatidylinositol 3-kinase (PI3K) (**E**,**F**), and extracellular signal-regulated kinase 1/2 (ERK1/2) (**G**,**H**) in HT-29 cells (*KRAS* WT CRC) and HCT-116 cells (*KRAS* MT CRC) according to a qPCR. Data are presented as the mean ± SD for two or three independent experiments performed in triplicate. * *p* < 0.05, ** *p* < 0.01 and *** *p* < 0.001 compared with the untreated control; ^##^
*p* < 0.01 ^###^
*p* < 0.001 compared with heteronemin.

**Figure 8 marinedrugs-20-00482-f008:**
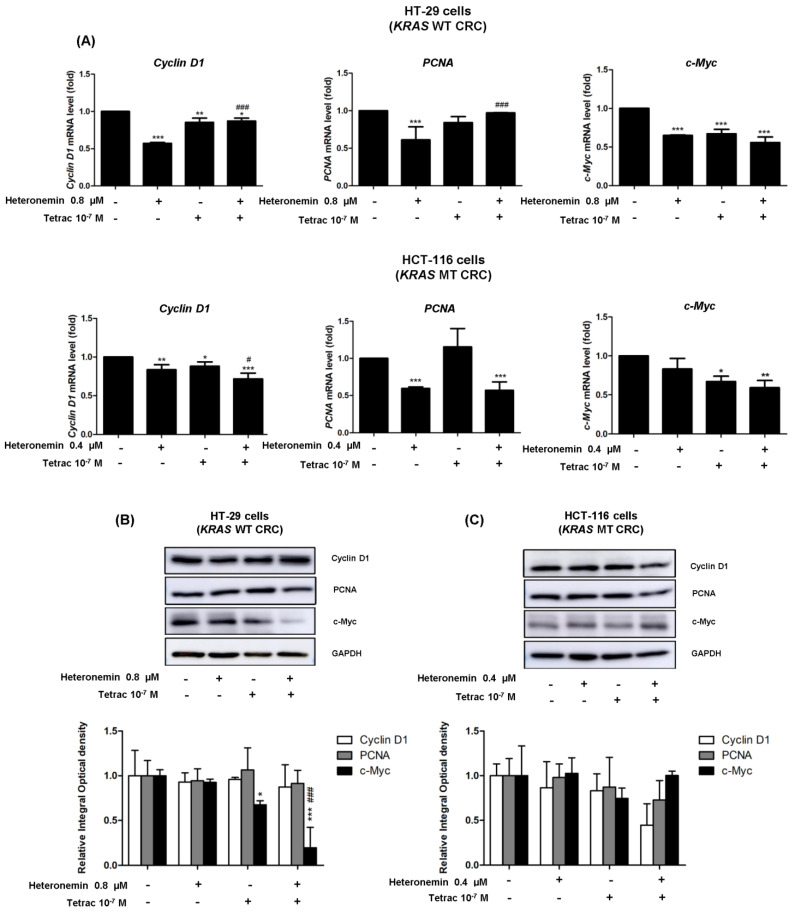
The combined treatment of heteronemin and tetrac alters the mRNA and protein expression of cell cycle regulators. Cells were treated with heteronemin, tetrac, and the combination for 24 h. Cells were harvested and total RNA was extracted for qPCR analyses of *cyclin D1*, proliferating cell nuclear antigen (*PCNA*), and *c-Myc* (**A**). HT-29 cells (*KRAS* WT CRC; upper panel) and HCT-116 cells (*KRAS* MT CRC; lower panel). Meanwhile, total proteins were extracted and Western blot analyses were conducted in HT-29 cells (**B**) and HCT-116 cells (**C**). Data are presented as the mean ± SD of three independent experiments. * *p* < 0.05, ** *p* < 0.01, and *** *p* < 0.001 compared with the untreated control; ^#^
*p* < 0.05 and ^###^
*p* < 0.001 compared with heteronemin.

**Figure 9 marinedrugs-20-00482-f009:**
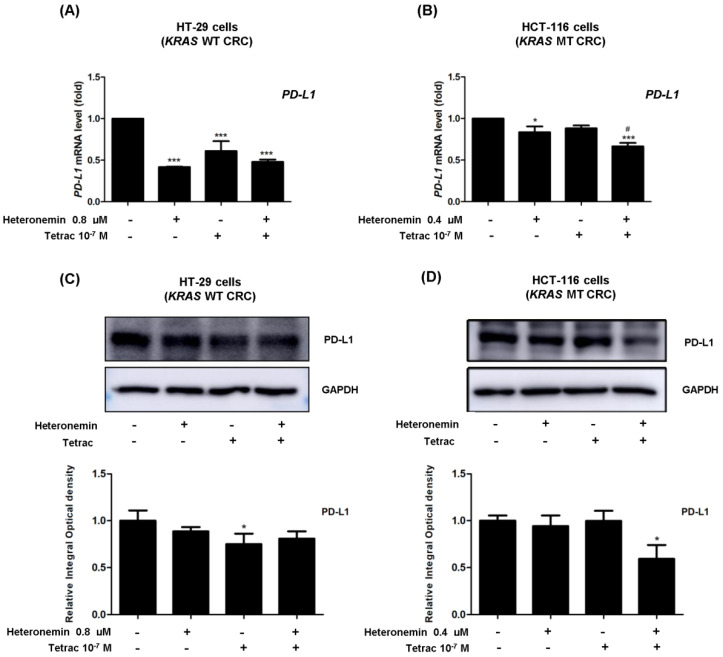
Tetrac with or without heteronemin alters the expression levels of PD-L1. RT-qPCR and Western blot analyses showed changes in mRNA levels (**A**,**B**) and protein levels (**C**,**D**) for 24 h in HT-29 cells (*KRAS* WT CRC) and HCT-116 cells (*KRAS* MT CRC). GAPDH was used as a loading control for RNA and protein normalization. Data are presented as the mean ± SD of three independent experiments. * *p* < 0.05 and *** *p* < 0.001 compared with the untreated control; ^#^
*p* < 0.05 compared with heteronemin.

**Table 1 marinedrugs-20-00482-t001:** Significant modulation of genes induced by heteronemin, tetrac, and their combination versus the control in HCT -116 cells (*KRAS* MT CRC).

Gene Name	Het vs. Ctrl	Gene Name	Het vs. Ctrl	Gene Name	Tet vs. Ctrl	Gene Name	Het + Tet vs. Ctrl	Gene Name	Het + Tet vs. Ctrl
*ACVR1C*	2.48	*ADAM9*	−1.40	*AP1M2*	−1.15	*ADAM17*	1.55	*AGGF1*	−1.20
*ACVRL1*	1.81	*AKAP2*	−1.10	*DPYSL3*	−1.12	*ADM2*	2.02	*AKT1*	−1.37
*ADAM17*	1.32	*AKT1*	−1.45	*LAMA5*	−1.10	*ANGPTL2*	1.56	*AP1M2*	−1.40
*ADM2*	1.97	*AKT2*	−1.32	*MAPK3*	−1.23	*ANGPTL4*	3.19	*ARAP2*	−1.58
*AMH*	1.71	*AP1M2*	−1.50	*OCLN*	−1.13	*BTG1*	1.63	*ATPIF1*	−1.34
*ANGPTL4*	2.16	*ARAP2*	−1.57			*CCDC80*	2.17	*BMP4*	−2.50
*ANPEP*	2.54	*ATPIF1*	−1.39			*CHRNA7*	1.76	*BMP7*	−1.27
*ARHGDIB*	2.42	*BMP4*	−2.30			*CLDN7*	1.31	*BMPR2*	−1.77
*CADM1*	2.47	*BMP7*	−1.45			*CLU*	2.01	*CALD1*	−1.51
*CCDC80*	2.61	*BMPR2*	−1.51			*COL7A1*	2.10	*CAMK2D*	−1.54
*CCL11*	2.49	*CALD1*	−1.47			*CTSL*	1.91	*CAV1*	−1.91
*CD34*	2.38	*CAMK2D*	−1.59			*CXCL8*	13.62	*CD2AP*	−1.24
*CEACAM6*	2.45	*CAV1*	−2.04			*CYB561*	1.13	*CD44*	−1.19
*CHRNA7*	1.70	*CD24*	−1.49			*F3*	1.38	*CD46*	−1.45
*CKMT1A*	2.70	*CD2AP*	−1.25			*FGF2*	1.64	*CDC42*	−1.27
*CLDN3*	1.32	*CD46*	−1.59			*FUT3*	1.54	*CEP170*	−1.34
*CLU*	1.59	*CDC42*	−1.34			*GDF15*	1.98	*CEP295*	−1.17
*COL1A1*	2.15	*CDH1*	−1.36			*HAS1*	1.37	*COL18A1*	−1.52
*COL5A2*	1.89	*CDKN1A*	−1.37			*HKDC1*	3.16	*COL6A1*	−1.38
*COL7A1*	2.17	*CEP295*	−1.36			*HMOX1*	2.50	*COL6A2*	−1.08
*COMP*	2.41	*CREBBP*	−1.21			*JUN*	4.77	*CREBBP*	−1.25
*CSPG4*	1.57	*CRIP2*	−1.39			*KRT7*	1.78	*CRIP2*	−1.52
*CTSK*	1.65	*CTNNB1*	−1.46			*LAMA3*	2.48	*CTNNB1*	−1.43
*CXCL8*	8.16	*CTNND1*	−1.49			*LAMA5*	1.12	*CTSH*	−2.07
*DPT*	2.73	*CTSH*	−2.21			*LAMB3*	3.57	*DAG1*	−1.40
*ELK3*	1.23	*DAG1*	−1.35			*LAMC2*	2.07	*DICER1*	−1.17
*EPN3*	1.73	*DICER1*	−1.18			*LLGL2*	1.14	*DPYSL3*	−1.41
*FUT3*	2.03	*DPYSL3*	−1.40			*MCAM*	1.26	*DST*	−1.26
*GJA5*	2.35	*DSC2*	−1.35			*MISP*	1.65	*EGFR*	−1.24
*HAS1*	2.14	*DST*	−1.34			*MYCL*	2.88	*EIF4E2*	−1.21
*HKDC1*	3.34	*EGFR*	−1.30			*NDRG1*	1.93	*ENPEP*	−1.32
*HMOX1*	2.14	*EIF4E2*	−1.18			*NRP1*	3.52	*ETV4*	−1.39
*ICAM1*	1.54	*ENPEP*	−1.35			*PDGFA*	1.34	*EVPL*	−1.50
*IL15*	1.91	*EP300*	−1.19			*PLAUR*	1.43	*FBLN1*	−1.52
*IL1A*	1.61	*ETV4*	−1.28			*POPDC3*	3.04	*FGF9*	−1.57
*IL1B*	2.57	*F11R*	−1.31			*PTPRM*	1.92	*FGFR3*	−1.40
*JUN*	3.51	*FN1*	−1.41			*RAC2*	1.17	*FN1*	−1.41
*KRT7*	2.46	*GALNT7*	−1.54			*SERPINE1*	1.81	*FSTL1*	−1.3
*LAMA3*	2.51	*GSN*	−1.67			*SH2B3*	1.14	*GALNT7*	−1.52
*LAMB3*	2.12	*GTF2I*	−1.59			*SH2D3A*	1.45	*GPI*	−1.28
*LAMC2*	1.79	*HIF1A*	−1.28			*SNAI1*	2.93	*GSN*	−1.90
*LLGL2*	1.10	*HIPK2*	−1.15			*SPP1*	3.78	*GTF2I*	−1.62
*LOXL2*	1.99	*HSD17B12*	−1.32			*THBS1*	1.53	*ID1*	−1.55
*MISP*	1.63	*ID1*	−1.60			*VEGFA*	1.67	*ID2*	−2.37
*MYCL*	3.23	*ID2*	−2.18					*ITGA2*	−1.32
*NDRG1*	1.50	*ITGA1*	−1.40					*ITGB4*	−1.67
*NR4A3*	1.58	*ITGA2*	−1.59					*JAG1*	−2.36
*NRP1*	3.26	*ITGA5*	−1.21					*KDM1A*	−1.54
*OVOL2*	1.80	*ITGB4*	−1.60					*KIAA1462*	−1.69
*PCOLCE*	2.49	*ITGB8*	−1.50					*LDHA*	−1.53
*PDGFA*	1.37	*JAG1*	−2.45					*LGALS1*	−1.34
*PFKFB1*	1.52	*KDM1A*	−1.63					*LTBP4*	−1.26
*POPDC3*	2.51	*KIAA1462*	−1.54					*MAPK3*	−1.71
*PRSS8*	2.00	*LDHA*	−1.52					*MTA1*	−1.18
*PTGS2*	2.55	*LGALS1*	−1.46					*MTBP*	−1.22
*PTK2B*	1.41	*MAP3K7*	−1.16					*MYO1D*	−1.31
*PTK6*	2.12	*MAPK3*	−1.72					*NFAT5*	−1.66
*PTPRM*	1.67	*MMP14*	−1.52					*NME1*	−1.39
*PTRF*	1.88	*MTA1*	−1.30					*NME4*	−1.48
*SDC4*	1.17	*MTOR*	−1.12					*PDGFC*	−1.22
*SERPINE1*	1.33	*MYO1D*	−1.26					*PIK3R2*	−1.37
*SH2D3A*	1.35	*NFAT5*	−1.55					*PLCG1*	−1.33
*SHB*	1.79	*NME1*	−1.48					*PLS1*	−1.24
*SLC37A1*	1.83	*NME4*	−1.22					*PNPLA6*	−1.13
*SLPI*	2.03	*P3H1*	−1.34					*PTTG1*	−1.56
*SNAI1*	2.31	*PGK1*	−1.49					*RAMP1*	−1.50
*SPDEF*	2.11	*PIK3CA*	−1.25					*RB1*	−1.30
*SPP1*	3.77	*POPDC3*	2.51					*RBL1*	−1.46
*SRPX2*	1.62	*PPP2R1A*	−1.45					*RBX1*	−1.30
*TACSTD2*	2.45	*PRKCZ*	−1.12					*RGCC*	−1.93
*TBXA2R*	1.95	*PTEN*	−1.32					*ROCK2*	−1.52
*THBS1*	1.49	*PTTG1*	−1.68					*SACS*	−1.16
*THY1*	2.06	*PXDN*	−1.63					*SCNN1A*	−1.51
*TNFSF13*	1.67	*RB1*	−1.27					*SET*	−1.28
*TYMP*	1.46	*RBL1*	−1.76					*SMAD1*	−1.32
*WARS*	1.55	*RBL2*	−1.20					*SMAD2*	−1.30
		*RGCC*	−1.55					*SMC3*	−1.27
		*ROCK2*	−1.50					*SNRPF*	−1.25
		*RPS6KB1*	−1.18					*SOD1*	−1.23
		*SACS*	−1.29					*SOX9*	−1.51
		*SCNN1A*	−1.74					*ST14*	−1.56
		*SERINC5*	−1.37					*STAT1*	−1.18
		*SET*	−1.23					*SYK*	−1.48
		*SLC2A1*	−1.19					*TFDP1*	−1.46
		*SMAD1*	−1.39					*TGFB1*	−1.31
		*SMAD4*	−1.14					*TGFBR2*	−1.21
		*SMC3*	−1.37					*TJP3*	−1.43
		*SMURF2*	−1.34					*TMC6*	−1.40
		*SNAI1*	2.31					*TSPAN1*	−1.63
		*SOD1*	−1.12					*VAMP8*	−1.34
		*SOD1*	−1.12					*VCAN*	−1.68
		*SORD*	−1.27					*WWTR1*	−1.20
		*SOX9*	−1.48						
		*ST14*	−1.55						
		*STAT3*	−1.28						
		*TCF20*	−1.29						
		*TFDP1*	−1.48						
		*TGFBR2*	−1.29						
		*TIMP1*	−1.45						
		*TJP3*	−1.13						
		*TMC6*	−1.49						
		*TNFSF12*	−1.28						
		*TP53*	−1.47						
		*TSPAN1*	−1.53						
		*VAMP8*	−1.59						
		*WARS*	1.55						
		*WWTR1*	−1.20						

**Table 2 marinedrugs-20-00482-t002:** Genes and growth factor pathways that drive multistep colorectal cancer progression.

Gene Name	Het vs. Ctrl	*p*-Value of Het vs. Ctrl	Tet vs. Ctrl	*p*-Value of Het vs. Ctrl	Het + Tet vs. Ctrl	*p*-Value of Het vs. Ctrl
*APC*	1.03	0.80	1.05	0.68	−1.05	0.58
*CTNND1*	−1.49	0.06	−1.23	0.37	−1.36	0.13
*EGFR*	−1.30	0.04	−1.08	0.29	−1.24	0.01
*IGF1*	1.79	0.19	1.16	0.42	1.21	0.42
*KRAS*	−1.20	0.08	−1.11	0.37	−1.11	0.10
*PIK3CA (PI3K*)	−1.25	0.03	−1.07	0.42	−1.15	0.08
*PTEN*	−1.32	0.02	−1.11	0.48	−1.17	0.06
*SMAD4*	−1.14	0.04	−1.09	0.25	−1.11	0.11
*TGFB1 (TGF-β*)	−1.11	0.06	−1.04	0.30	−1.31	0.02
*TGFB2 (TGF-β*)	−1.11	0.06	−1.09	0.42	−1.09	0.42
*TGFBR2*	−1.09	0.43	1.07	0.54	−1.21	0.04
*TP53*	−1.47	0.04	−1.10	0.46	−1.39	0.01
*VEGFA*	1.43	0.06	1.01	0.89	1.67	0.01
*VEGFB*	−1.14	0.44	−1.07	0.40	−1.28	0.18

**Table 3 marinedrugs-20-00482-t003:** Gene primer sequences in this study.

Primer Name	Forward Primer (5′-3′)	Reverse Primer (5′-3′)
*EGF*	TTCTGTGGGAGCAGTGTGA	CCTCACCACCACAGGTTTCT
*EGFR*	AATTTACAGGAAATCCTGCATGGC	GATGCTCTCCACGTTGCACA
*ERK1/2*	GCAGCTGAGCAATGACCATA	TTGCTTGCAATGTTCTCCTG
*PI3K*	CCTGATCTTCCTCGTGCTGCTC	ATGCCAATGGACAGTGTTCCTCTT
*PD-L1*	GTTGAAGGACCAGCTCTCCC	ACCCCTGCATCCTGCAATTT
*CCD1*	CAAGGCCTGAACCTGAGGAG	GATCACTCTGGAGAGGAAGCG
*PCNA*	TCTGAGGGCTTCGACACCTA	TCATTGCCGGCGCATTTTAG
*c-Myc*	TTCGGGTACTGGAAAACCAG	CAGCAGCTCGAATTTCTTCC
*GAPDH*	AGGGCTGCTTTTAACTCTGGT	CCCCACTTGATTTTGGAGGGA

## Data Availability

The original contributions presented in the study are included in the article; further inquiries can be directed to the corresponding author.

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
