# Peer review of "Heteronemin and Tetrac Induce Anti-Proliferation by Blocking EGFR-Mediated Signaling in Colorectal Cancer Cells"

_marinedrugs, 2022, doi:10.3390/md20080482_

Round 1

Reviewer 1 Report

The authors have investigated the effects of heteronemin and tetrac on colorectal cancer cells.  This builds on previous work by the group looking at the effects of these compounds on small cell lung cancer cells.  They have compared the actions of the compounds in HT-29 cells with wild type KRAS and HCT-116 cells with mutant KRAS.

Comments:

With the use of the two different colorectal cancer cell lines, consistency through the text would be helpful – in some places the HT-29 and HCT-116 cells are discussed but in other places the cell lines are discussed in the context of KRAS status.  It would aid the narrative to put the name of the cell line and then in brackets the KRAS status in all locations, including figure legends.

Figure 5 – please further discuss the clustering of the treatment groups in the heatmap, as one of the tetrac repeats clusters with the heteromenin repeats and two tetrac repeats cluster with the control repeats.

Line 281-282, look at phrasing to ensure meaning is clear.

Figure 7 – (D) needs a label on the figure.

Lines 288-289 – PI3K mRNA levels are said to show a decreasing trend in the colorectal cancer cells following treatment with the compounds but treatment of HT-29 cells with both compounds together does not show a decrease in PI3K mRNA levels compared to the control.  Please clarify.

Figure 8 – please comment on the possibility of increased PCNA mRNA levels with tetrac treatment of HCT-116 cells.

Please explain why the Nanostring nCounter technology approach was only used for the HCT-116 cells and not also the HT-29 cells.  All other experiments have compared the data for the two cell lines.

Methods section – antibody concentrations should be cited for the Western blot analysis and primer concentrations should be cited for the RT-qPCR section.

Reviewer 2 Report

The present study is well described with sufficient proofs that supports the hypothesis given in the introduction.
The reviewer recommends the acceptance of the present study in its current form but only after revising the reference list that needs a compete check to assure being applying the instructions included in the instructions for authors file on Marine Drugs website.

Round 2

Reviewer 1 Report

Thank you for your updates following the original comments.

The figure legends are now clearer with the addition of (KRAS WT CRC) after HT-29 cells and (KRAS MT CRC) after HCT-116 cells.  However, I feel the addition of these brackets/phrases would benefit the text as a whole and should be included after each mention of the two cell lines within the main body of the manuscript.

Figure 5 – I stand by my original comment that in Figure 5A one of the tetrac repeats clusters with the heteromenin repeats and two tetrac repeats cluster with the control repeats.  I understand that Figure 5B is the mean of each treatment in the heat map.  However, a comment could be included in the figure legend in line with the response that has been provided to the original comment.

Line 281-282, look at phrasing to ensure meaning is clear.  This comment has not been responded to – the original sentence was “When KRAS WT cells were treated with heteronemin alone significantly reduced levels of EGF and EGFR mRNA compared to the control …”.  Suggestion that this should be corrected to: “When KRAS WT cells were treated with heteronemin alone, this significantly reduced levels of EGF and EGFR mRNA compared to the control…”.
